# Morphological Variation of the Mandible in the Orthognathic Population—A Morphological Study Using Statistical Shape Modelling

**DOI:** 10.3390/jpm13050854

**Published:** 2023-05-19

**Authors:** Hylke van der Wel, Bingjiang Qiu, Fred K. L. Spijkervet, Johan Jansma, Rutger H. Schepers, Joep Kraeima

**Affiliations:** 1Department of Oral and Maxillofacial Surgery, University Medical Center Groningen, Hanzeplein 1, 9713 GZ Groningen, The Netherlands; 2Department of Radiology, Guangdong Provincial People’s Hospital, Guangdong Academy of Medical Sciences, Guangzhou 510080, China; 3Guangdong Provincial Key Laboratory of Artificial Intelligence in Medical Image Analysis and Application, Guangdong Provincial People’s Hospital, Guangdong Academy of Medical Sciences, Guangzhou 510080, China; 4Department of Oral and Maxillofacial Surgery, Expertcenter for Orthofacial Surgery, Martini Hospital Groningen, Van Swietenplein 1, 9728 NT Groningen, The Netherlands

**Keywords:** orthognathic surgery, virtual surgical planning, statistical shape model, cephalometry

## Abstract

The aim of this study was to investigate the value of 3D Statistical Shape Modelling for orthognathic surgery planning. The goal was to objectify shape variations in the orthognathic population and differences between male and female patients by means of a statistical shape modelling method. Pre-operative CBCT scans of patients for whom 3D Virtual Surgical Plans (3D VSP) were developed at the University Medical Center Groningen between 2019 and 2020 were included. Automatic segmentation algorithms were used to create 3D models of the mandibles, and the statistical shape model was built through principal component analysis. Unpaired *t*-tests were performed to compare the principal components of the male and female models. A total of 194 patients (130 females and 64 males) were included. The mandibular shape could be visually described by the first five principal components: (1) The height of the mandibular ramus and condyles, (2) the variation in the gonial angle of the mandible, (3) the width of the ramus and the anterior/posterior projection of the chin, (4) the lateral projection of the mandible’s angle, and (5) the lateral slope of the ramus and the inter-condylar distance. The statistical test showed significant differences between male and female mandibular shapes in 10 principal components. This study demonstrates the feasibility of using statistical shape modelling to inform physicians about mandible shape variations and relevant differences between male and female mandibles. The information obtained from this study could be used to quantify masculine and feminine mandibular shape aspects and to improve surgical planning for mandibular shape manipulations.

## 1. Introduction

Orthognathic surgery is a complex surgical procedure that involves repositioning of the jaws to correct a dysfunctional occlusion. While functional improvement is the main goal, the optimization of facial harmony and aesthetics has also become an important motivation for undertaking orthognathic surgery [1]. The emergence of 3D models and virtual surgical planning plays an important role because, when combined with cephalometry, it allows for facial harmonization during the treatment planning instead of just relying on the facial profile seen on the traditional 2D images [2,3]. Two important determinants in facial aesthetics are facial symmetry and perceived sexual dimorphism [4], and the shape and soft tissue projection of the mandible play an important role here. Regarding facial asymmetry, the probability that this is due to a mandibular deformity is higher than due to a maxillary deformity [5]. Mandibular shape changes are also an important part of feminization and masculinization surgery involving contouring of the angles, basilar rims, and genioplasty [6]. Correcting a mandibular deformity during orthognathic surgery needs a specific pre-operative planning design or surgical template for a planned outcome [5,6].

State of the art pre-operative planning is often based on a combination of cephalometric measurements and soft-tissue simulations. Cephalometric measurements guide the pre-operative situation towards normal values, e.g., by measuring the cant of the occlusion plane and adjusting it virtually. Next, a 3D soft-tissue simulation informs the surgeon about any expected changes in soft-tissue profile. As noted, the cephalometric planning is almost always revised after a 3D surgical simulation [3]. Apparently, after viewing the 3D virtual model on the computer screen, surgeons and orthodontists change their opinion about the treatment plan. We know that in clinical practice, there will be a difference in pre-operative planning and design based on the surgeon involved, as they have their own measurement methods and preferred references [2]. Even though cephalometric measurements can quantify and objectify the pre-operative planning, the subjective judgement of the changes in 3D profile seems to have a considerable influence on the final pre-operative design. However, although virtual 3D planning and simulations enable the visualization of countless different operations, there does not seem to be a consensus on the normal values. 

A more objective and quantifiable approach to assess 3D shape changes is currently lacking from the literature but, if developed, could improve orthognathic surgery planning and support the data driven decision making. Hence, this study focused on the mandible by investigating the feasibility of an orthognathic surgery planning approach based on the statistics of a population’s 3D mandibular shape. A further aim was to present an objective way to determine how masculine or feminine a mandible of a given patient is, relatively, based on the proposed population statistics method for sexual dimorphism and symmetry.

## 2. Materials and Methods

### 2.1. Study Population

The pre-operative CBCT scans of all the patients for whom a 3D Virtual Surgical Plan (3D VSP) was made, for either a Bilateral Sagittal Split Osteotomy (BSSO) or bimaxillary surgery at the University Medical Center Groningen in the years 2019 and 2020, were considered for inclusion. Patients were excluded if the mandible was not fully within the field-of-view, or the mandible had been operated on before. The age at scanning and sex was collected from the DICOM header info.

### 2.2. Generation of 3D Models

Image segmentation of the mandibles was performed with an in-house developed and validated automatic segmentation algorithm [7], after which all the mandibles were inspected visually to make sure that the segmentation had been performed correctly. If not, improvements were made manually via Mimics Medical (Mimics Medical, Materialise, Leuven, Belgium) by one of the authors (HvdW). The 3D mandible models were re-meshed to a vertex edge-length of 1.0 mm to determine their resolution. 

### 2.3. Statistical Shape Model

To create a Statistical Shape Model (SSM), the digital 3D models of the mandibles need to be in so-called landmark-correspondence, meaning that all the landmarks have to match each other. Landmark-correspondence between all the mandibles was achieved by registering a template to each mandible using a registration algorithm implemented in MATLAB (MathWorks, Natick, MA, USA) [8]. A dataset was created of evenly distributed landmarks over mandible surfaces. Each landmark’s mean position was calculated to determine the mean mandible model. Using Principal Component Analysis, the principal components (PCs) of shape variation from the mean model were calculated. Figure 1 shows the statistical shape modelling workflow. The shape variation of each PC could be visualized by first plotting the mean shape, and the ±3 standard deviation (SD) in shape, to account for 99.7% of the population. Comparing the resulting images with the mean shape provided an insight into each PC’s anatomical variation.

### 2.4. Evaluation of the Statistical Shape Model

The SSM should be efficient in its description of the shape variation through a minimal number of PCs. A compactness graph was made from the SSM to study how many PCs are needed to describe 95% of the shape variation [9]. To test if we had included enough mandibles to describe the entire orthognathic population, a leave-on-out-cross-validation experiment was performed with 3 mandibles. The results were plotted as a generalization graph to determine if the current study’s results could be generalized to the entire population [9,10].

### 2.5. Difference between Male and Female Mandibles

A test for statistically significant differences between male and female orthognathic mandibular shapes was set up. Each individual mandible in the SSM was described by its variation from the mean model per PC. Therefore, an individual mandible was described by a list of relative weighting factors for each PC; the mean mandible had a value of 0 for all PCs. If, for example, a model varied 2 standard deviations from the mean in the first principal component, its weighting factor for the first PC was +2. We used an unpaired *t*-test to check for significant differences (*p* < 0.05) in mandible shape weighting factors between males and females [11].

## 3. Results

### 3.1. Inclusion and Evaluation of the Statistical Shape Model

A total of 194 patients was included in this study. Their mean age was 24.4 ± 8.5 years, and 130 of the included patients were female. A statistical shape model was created with 193 principal components, describing 100% of the shape variation. A compactness graph showed that the first 80 PCs in this model described 95% of the shape variation (see Figure 2). The first (13.8%), second (8.1%), and third (6.7%) components described relatively large parts of the variation, after which the curve flattened out and the subsequent PCs described less than 5% of the variation. An evaluation of how generalizable the SSM is resulted in the generalization curve in Figure 3.

### 3.2. Orthognathic Population Shape Variation

The variations between all 193 PCs were visualized and reviewed, and the first 5 PCs were selected because they demonstrated most of the variance and displayed relevant anatomical variation. Figure 4 shows these first 5 PCs of orthognathic mandibular shape variation. The first and, therefore, most prevalent component of variation mainly describes the height of the vertical dimensions of the mandibular ramus and condyles. The second component mainly describes variation in the gonial angle of the mandible. The third component describes the width of the ramus and the anterior/posterior projection of the chin quite well. The fourth principal component mainly describes the lateral projection of the angle of the mandible. The fifth descriptive principal component of variation seems to be mainly the lateral slope of the ramus and the inter-condylar distance, varying between a ‘V-shaped’ mandible and a ‘Rectangular-shaped’ mandible.

### 3.3. D VSP Based on the Principal Components of Orthognathic Shape Variation

Statistically significant differences were found in some of the PCs of shape variation between male and female mandibles in the SSM. Table 1 shows the weighting factors (mean ± SD) of the PCs with a significant difference. Since the weighting factors were on a scale from −3 to +3 SD from the mean shape, the resulting mean differences also represented a clinically significant difference in shape. This difference was used to illustrate the possibility of a patient specific 3D VSP based on a SSM analysis of the population. The mean mandible shape of the SSM is depicted in Figure 5. Using the findings presented in Table 1, the relevant weighting factors of this mean male mandible were changed to the mean weighting factors of the female mandible; these are presented in Figure 5. Changing the relevant weighting factors along the continuous −3 to +3 SD scale changed the shape of the mandible. Based on the found relationship between the ‘Male/Female’ variable inputs and the shape of the mandible, the shape of the male mandible input could be changed towards a more likely female shape while still staying within the probable shape of the original male mandible.

## 4. Discussion

In this study, the 3D mandible shapes of 194 orthognathic patients were analysed to find orthognathic mandible shape variations. The first orthognathic mandible SSM is visualized in Figure 4, with a description of the most important shape variations. It is feasible to use the SSM based planning to correct for sexual dimorphism in mandibular shape. 

The SSM-based approach presented above for orthognathic surgery planning has not been reported before, although several studies have investigated the potential of using a SSM based approach for defect reconstructive surgery of the mandible [12,13,14]. With the shape information from as few as 11 healthy mandibles, Zachow et al. demonstrated it is possible to propose a reasonable reconstruction shape for a deformed mandible [14]. Expanding on these first results, two studies noted the possibility of reconstructing even larger corpus defects [12,13]. This previous research paved the way for the feasibility of building a SSM of the mandible and applying it to generate a virtual planning template.

Morphometric measurement of the mandible has, in the previous literature, often been done using cephalometric methods. In a publication by Vallabh et al., a link between cephalometric measurements and the entire three-dimensional morphology of the mandible was studied using statistical shape modelling [15]. They selected, from the SSM built from 65 edentulous mandibles, seven cephalometric measurements, namely, the pogonion-lateral condylar distance, ramus height, intercondylar distance, ramus width, pogonion-interdental distance, and body length, as input variables to obtain the most accurate shape prediction of the entire mandible morphology. Compared to their work, the current results, which concentrated on orthognathic surgery, involved the addition of soft-tissue cephalometric measurements to the mandibular shape, thereby enhancing the SSM based orthognathic surgery planning approach with a soft-tissue prediction.

An in-house developed method for automatic segmentation of mandible CT data was used to create the dataset for this study [7]. We show that it is possible to automatically place numerous landmarks onto the segmented mandible with the registration method described in this article. Previous research already pointed out that automatic landmarking is advantageous for cephalometric analyses [16].

It is apparent from the compactness experiment performed in this study that 80 PCs are necessary to describe 95% of the shape variation in the population. Compared to SSMs of other anatomical structures, 80 seems to be a lot. We suppose that this is due to two factors: the complexity of the mandibular shape and the quality of the segmented mandibles. First of all, the highly complex mandible demonstrates a lot of inter-person variation. Secondly, the segmentations used to build the SSM were made from CBCT scans containing orthodontic braces. This results in segmentations with a lot of small variations around the teeth, which is generally the case for this area. Ultimately, a compact model seems preferable for 3D VSP, which suggests that future models should reduce the level of detail in the dental area.

Our study focused on the mandible shape in an orthognathic population. An elaboration of the current data with a large dataset of non-orthognathic mandibles will offer the possibility of comparing an orthognathic mandible with the normal values of shape variation within the population. The composition of the population within the statistical shape model is very important. Generalization experiments, such as those performed in this study, should inform us if the SSM represents the population accurately. Generally, the population of the SSM needs to be suitably large so as to not be affected by, for example, individual variations such as Stafne bone cavities that are prevalent in 0.17% of the population [17]. Furthermore, the distribution of such variables as age and patient sex should be monitored. Although the age distribution in this study (24.4 ± 8.5) is representative of our orthognathic population, the known effects of mandible morphology changes with age were not studied separately due to the small age distribution. The imbalance between the number of male and female patients in this study is representative of our population.

Using a data-driven approach to find the desired surgical correction-values is not the standard of care yet in orthognathic surgical planning. It was not this study’s aim to replace current clinical observations and 3D cephalometric analyses for making a surgical plan. However, the authors believe that, as a concept, the SSM approach can support the decision making and planning process in orthognathic surgery in order to improve the workflow and make it less user dependent.

This study presents the feasibility of an orthognathic surgery planning approach based on statistical shape modelling. The most important aspects of shape variation in our orthognathic population were described and displayed. Studying the entire 3D morphology variation could add a new perspective to 3D VSP, as was exemplified by analysing and manipulating sex-related principal components.

## Figures and Tables

**Figure 1 jpm-13-00854-f001:**
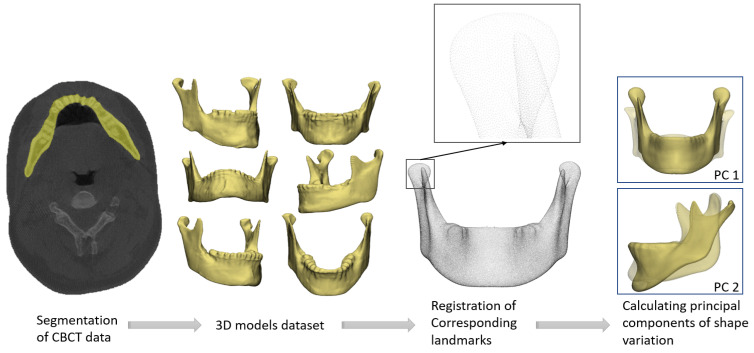
Approach to create a Statistical Shape Model from patients’ CBCT data. First, semi-automatic segmentation of the CBCT data was performed to create 3D mandible models. After this, all the mandible models were registered, ensuring they contained the same corresponding landmarks on their surfaces. The variation in landmark positions between all the mandibles was used to calculate the principal components of shape variation. The first two components are shown in this figure.

**Figure 2 jpm-13-00854-f002:**
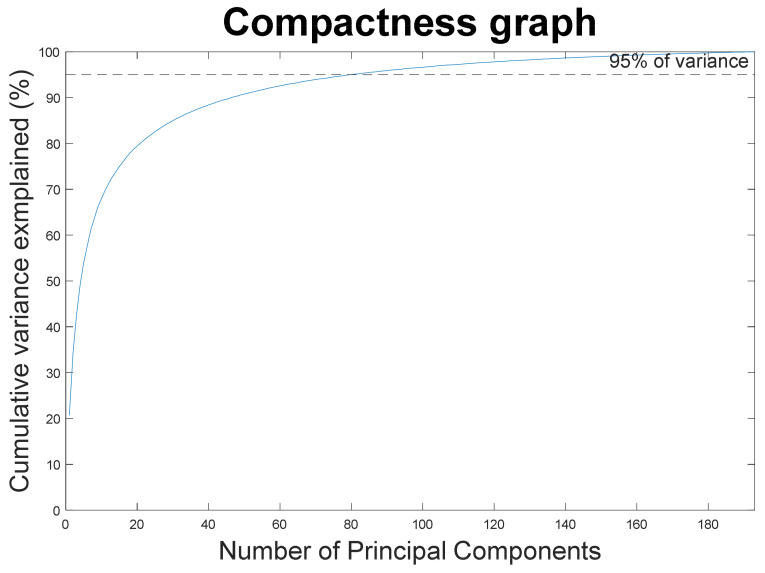
A SSM compactness graph to explain the components after plotting the number of PCs against the cumulative variance. A total of 193 PCs were used in the SSM which, combined, explained 100% of the variance. Only the first 80 PCs were necessary to describe 95% of the variance (dotted line).

**Figure 3 jpm-13-00854-f003:**
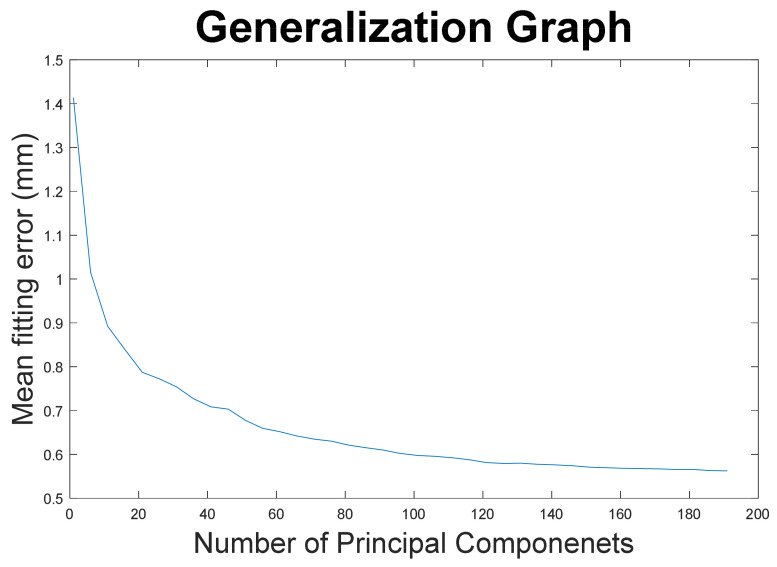
The SSM generalization graph shows how well the model will fit a new orthognathic mandible that is currently not in the SSM. The shown mean curve for 3 leave-one-out experiments results in a mean error of 0.56 mm on applying all the PCs.

**Figure 4 jpm-13-00854-f004:**
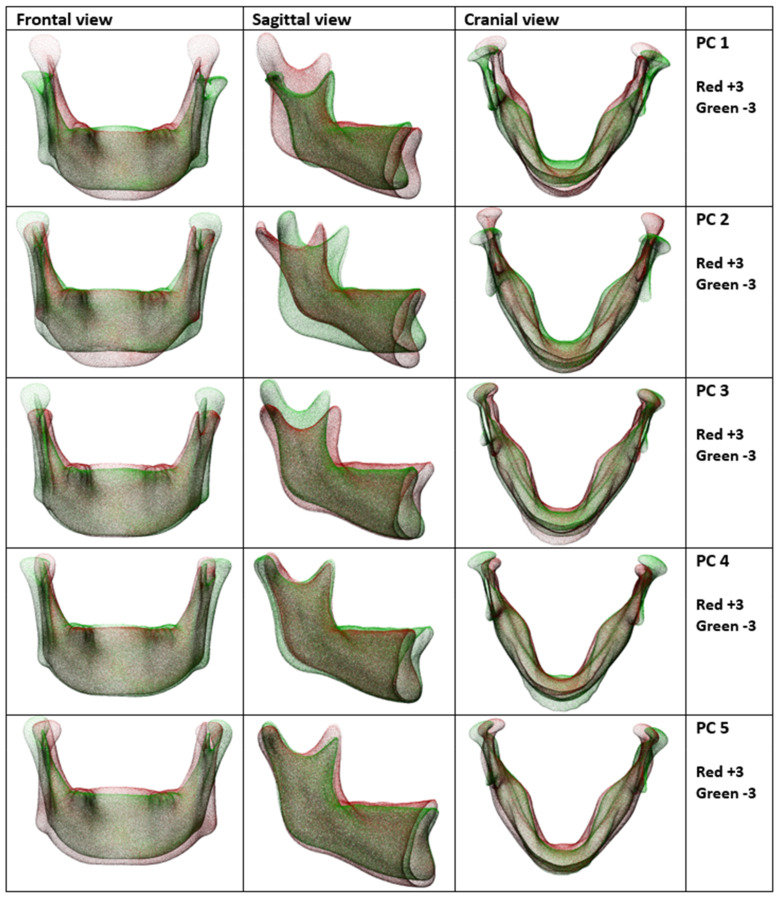
A visualization of the most influential PCs in orthognathic mandibular shape variation. Combined, these five PCs describe 55% of the total shape variation. The variation between the +3 and -3 standard deviations from the mean is shown for each PC, adding up to a 97% variation among the population.

**Figure 5 jpm-13-00854-f005:**
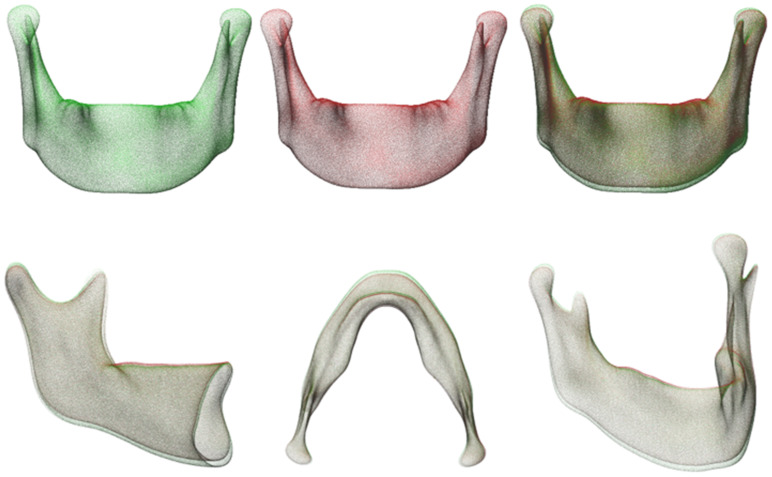
An example of a 3D VSP based on the analysis of principal component shape variation. Top left in green: the mean male mandible shape. In red: the same male mandible but with the weighting factors of the relevant PCs changed slightly to the mean female weighting factors. The top right image and the lower three images depict the differences in contour after overlaying both mandibles.

**Table 1 jpm-13-00854-t001:** The mean values and standard deviations of the male and female population in the SSM with a significant difference in PCs.

PC	Mean Male Weighting Factor	Mean Female Weighting Factor	Mean Difference	*p*-Value
1	0.23 ± 0.89	−0.11 ± 1.03	0.34	0.026
3	−0.33 ± 0.86	0.01 ± 0.98	0.49	<0.001
5	0.41 ± 1.02	−0.21 ± 0.93	0.61	<0.001
6	−0.34 ± 0.97	0.17 ± 0.98	0.50	<0.001
8	0.24 ± 0.92	−0.12 ± 1.02	0.36	0.017
13	0.38 ± 0.99	−0.18 ± 0.96	0.56	<0.001
20	−0.21 ± 0.95	0.10 ± 1.01	0.31	0.042
24	0.24 ± 1.07	−0.12 ± 0.94	0.36	0.018
29	−0.32 ± 1.01	0.16 ± 0.96	0.47	0.002
43	−0.23 ± 1.10	0.11 ± 0.93	0.33	0.029

## Data Availability

The data presented in this study are available on request from the corresponding author. The data are not publicly available due to data sharing agreements.

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
