# Peer review of "Morphological Variation of the Mandible in the Orthognathic Population—A Morphological Study Using Statistical Shape Modelling"

_jpm, 2023, doi:10.3390/jpm13050854_

Round 1
Reviewer 1 Report
The authors present interesting findings of what may be considered as future treatment planning. As digitalization becomes more and more important, that paper may be of interest to the readers of the journal. I have only one minor remark. The paper should undergo meticulous English review, as there are some minor grammatical and expression issues trhoughout the whole paper (e. g. lines 235-236 "We propose this is due to two factors..." - should it be suppose?)
After such a minor revision the paper may be considered for publication.
Author Response
Dear Reviewer,
We want to thank you for the feedback provided to us. As per your suggestion the manuscript is sent to a native English speaker for meticulous English review, before resubmitting it to the journal
Yours sincerely,
H. van der Wel,
On behalf of all authors.
Reviewer 2 Report
This is an interesting study that investigates the morphological variation of the mandible. The study was appropriately conducted. There are a few comments to improve the paper.
It is previously known that mandible morphology changes with age (https://www.ncbi.nlm.nih.gov/pmc/articles/PMC6131965/). Is it possible to analyze the results across different age groups?
How did the automatic 3D reconstruction deal with starburst artifact caused by dental filling?
The sentences in from line 137 to 139 must be removed.
The study had a much smaller number of male individuals. Why was that? The imbalance between the two sexes should be noted as a limitation.
Please add subsection numbers 2.1, 2.2, etc to all subsections in the methods section.
The authors should be aware of the Stafne defect, an asymptomatic bone defect that reduces the volume of the mandible. Was it found in any of the subjects? Its prevalence may be added to the discussion. https://www.sciencedirect.com/science/article/pii/S1991790222002057.
Line 10: correct the spelling of “China”
Author Response
Dear reviewer,
We want to thank you for the feedback we have received on our manuscript. Based on the helpful suggestions we have made some revisions and added a literature reference based on your suggestion. The full response can be found as attachment to this message.
Yours sincerely,
Hylke van der Wel,
On behalf of all authors,

Reviewer 3 Report
The manuscript titled " Morphological variation of the mandible in the orthognathic population – A morphological study using Statistical Shape Modelling is an original article focusing on shape variation of the mandible among male and female population. The article is well written and scientifically sound. The article needs some modifications to make it more interesting for the readers. Following are my suggestions to be incorporated in the manuscript,
1. The introduction needs to be modified. The authors haven’t mentioned clearly the statement of the problem like, why the two methods are more commonly used compared to other methods used in implant surgery and the rationale of comparing these 2 methods alone. Try to add a few more works of literature to support the statement.
2. The background of biomedical needs to be still elaborated for a better understanding of the existing conditions which need this approach.
3. 2-3 lines in the last paragraph stating the hypothesis of the research and what ways this article will stand alone from previously published literature on this topic.
4. What was the inclusion and exclusion criteria used for the sample.
5. How was the sample size determined?
Author Response
Dear reviewer,
May we thank you for the feedback we have received, based on your helpfull insights and feedback we have made some revisions to the manuscript. Please find attatched our detailed response to your suggestions,
Yours sincerely,
H. van der wel,
On behalf of all authors,

Round 2
Reviewer 3 Report
The manuscript is modified according to the mentioned comments.